# Sustainable Biomass Lignin-Based Hydrogels: A Review on Properties, Formulation, and Biomedical Applications

**DOI:** 10.3390/ijms241713493

**Published:** 2023-08-30

**Authors:** Chaymaa Hachimi Alaoui, Gildas Réthoré, Pierre Weiss, Ahmed Fatimi

**Affiliations:** 1Chemical Science and Engineering Research Team (ERSIC), FPBM, Sultan Moulay Slimane University, Mghila, P.O. Box 592, Beni Mellal 23000, Morocco; chaymaa.hachimi-alaoui@etu.univ-nantes.fr; 2Nantes Université, Oniris, Univ Angers, INSERM, Regenerative Medicine and Skeleton, RmeS, UMR 1229, F-44000 Nantes, France; 3Nantes Université, Oniris, Univ Angers, CHU Nantes, INSERM, Regenerative Medicine and Skeleton, RmeS, UMR 1229, F-44000 Nantes, France; gildas.rethore@univ-nantes.fr (G.R.); pierre.weiss@univ-nantes.fr (P.W.)

**Keywords:** lignin, chemistry, hydrogel, tissue engineering, regenerative medicine, 3D bioprinting

## Abstract

Different techniques have been developed to overcome the recalcitrant nature of lignocellulosic biomass and extract lignin biopolymer. Lignin has gained considerable interest owing to its attractive properties. These properties may be more beneficial when including lignin in the preparation of highly desired value-added products, including hydrogels. Lignin biopolymer, as one of the three major components of lignocellulosic biomaterials, has attracted significant interest in the biomedical field due to its biocompatibility, biodegradability, and antioxidant and antimicrobial activities. Its valorization by developing new hydrogels has increased in recent years. Furthermore, lignin-based hydrogels have shown great potential for various biomedical applications, and their copolymerization with other polymers and biopolymers further expands their possibilities. In this regard, lignin-based hydrogels can be synthesized by a variety of methods, including but not limited to interpenetrating polymer networks and polymerization, crosslinking copolymerization, crosslinking grafted lignin and monomers, atom transfer radical polymerization, and reversible addition–fragmentation transfer polymerization. As an example, the crosslinking mechanism of lignin–chitosan–poly(vinyl alcohol) (PVA) hydrogel involves active groups of lignin such as hydroxyl, carboxyl, and sulfonic groups that can form hydrogen bonds (with groups in the chemical structures of chitosan and/or PVA) and ionic bonds (with groups in the chemical structures of chitosan and/or PVA). The aim of this review paper is to provide a comprehensive overview of lignin-based hydrogels and their applications, focusing on the preparation and properties of lignin-based hydrogels and the biomedical applications of these hydrogels. In addition, we explore their potential in wound healing, drug delivery systems, and 3D bioprinting, showcasing the unique properties of lignin-based hydrogels that enable their successful utilization in these areas. Finally, we discuss future trends in the field and draw conclusions based on the findings presented.

## 1. Introduction

As a result of the increasing environmental effects caused by the fossil fuel industry, there has been an increased interest in finding alternative, clean, and globally available resources [1]. The use of lignocellulosic biomass materials can reduce dependence on petrochemical resources [2]. As evident from the literature, lignocellulosic biomass is found on top of agricultural waste, which is often likely to be disposed of by burning to generate energy [3,4,5]. This practice causes serious environmental problems [6]. Therefore, in recent years, lignocellulosic materials have attracted the special interest of researchers owing to their wide availability, biorenewable nature, and non-competitiveness with food [2,6]. From the environmental and economic points of view, they have proven to be good materials for the production of high-value-added products [7,8]. Lignocellulosic biomass mainly consists of 35–50 wt.% cellulose, 20–35 wt.% hemicellulose, and 10–25 wt.% lignin [9]. However, the contents of these three polymers depend on their origin, plant species, and environmental conditions [5,10]. Various processes have been developed to overcome the recalcitrant nature of lignocellulosic biomass, which is caused by the cellulose’s crystallinity, the lignin’s hydrophobicity, and the cellulose’s encapsulation by the lignin–hemicellulose network [9]. Lignin biopolymer is the principal recalcitrant component in lignocellulosic biomass, due to its very complicated structure [7], made up of propylphenolic subunits; therefore, its valorization into valuable materials is highly challenging [11]. Lignin has received a lot of interest from researchers in recent years. They have been studying the physicochemical behavior of lignin and its novel features in order to apply them to a variety of formulations [12]. Moreover, due to its key properties (e.g., biodegradability, thermal stability, reactivity, etc.), lignin is considered to be a good candidate for the development of advanced materials, including hydrogels, nanotubes, films, nanofibers, and nanoparticles, for a variety of applications [5,13,14,15,16].

Hydrogels are a category of soft materials that have received growing attention due to their unique properties, such as high water content, elasticity, flexibility, and biocompatibility [17,18]. In general, hydrogels can be used in a wide range of applications, such as hygiene, agricultural water retention, carbon capture, and biomedical applications [19]. The creation of such hydrogels has shifted from the adaptation of synthetic polymers to the chemical modification of biopolymers due to the latter’s biocompatibility, biodegradability, low toxicity, susceptibility to enzymatic degradation in most cases, and environmentally favorable characteristics [20]. Lignin, among other biopolymers, is a promising candidate for the development of novel hydrogels thanks to its characteristics, such as the exhibition of antioxidant and antimicrobial properties, an anti-inflammatory effect, biocompatibility, and low cytotoxicity [14,21,22]. In biomedical applications, lignin-based hydrogels have shown potential for uses in tissue engineering [23], wound healing [24], drug delivery [25], 3D bioprinting [26], and other applications [27,28]. For these applications, the potential of turning lignin into functional biomaterials such as hydrogels has been explored and is rapidly growing [29].

In most cases, for biomedical applications, lignin biopolymers have been used in different hydrogel formulations (based on synthetic polymers or biopolymers) to facilitate their formation and improve their biological and/or physicochemical properties. Therefore, it has been necessary to use copolymerization to ensure good water retention and high mechanical properties [30]. For example, it has been confirmed that the presence of lignin in the hydrogel system could potentially enhance the swelling and thermo-oxidative stability properties [31]. Furthermore, lignin-based hydrogels can be synthesized by a variety of methods, including chemical or physical interactions, and in both cases the lignin could be used either as a crosslinked unit for lignin hydrogels (e.g., phenol–lignin–formaldehyde [32], lignin–cellulose [33], lignin–polyurethane (PU) [24], etc.) or as a crosslinking agent for other hydrogels (e.g., acrylic acid [25], chitosan [23], etc.).

Lignin-based hydrogels have received a lot of interest in recent years. As a result, research and development, particularly on their formulation and use in pharmaceutical and biomedical applications, is growing significantly [34]. Nevertheless, few studies on the use of lignin-based hydrogels in tissue engineering, wound healing, drug delivery, and 3D bioprinting have been summarized as review reports and published during the past five years [27,35,36,37,38]. Given this need, we propose hereinafter a review of sustainable biomass lignin-based hydrogels for biomedical applications. The studies and results covered in this review are outstanding developments in molecular research and are from papers published within the last five years (above 50%).

In this review paper, we provide an in-depth analysis of the current state of research on lignin-based hydrogels and their biomedical applications. The first part presents an extensive review of lignin processing methods for extraction and isolation, highlighting the various techniques employed to obtain lignin from different sources. We delve into the composition and structure of lignin, examining the complex molecular arrangements that contribute to its unique properties. Part 2 focuses on lignin-based hydrogels, providing an overview of hydrogels. Through a comprehensive analysis of the literature, we explore the preparation methods used to create lignin-based hydrogels. Part 3 delves into the exciting biomedical applications of lignin-based hydrogels. We discuss their versatility and potential use in tissue engineering, where they can provide a biomimetic scaffold for cell growth and regeneration. Additionally, we explore their potential in wound healing, drug delivery systems, and 3D bioprinting, highlighting the unique properties of lignin-based hydrogels that enable their successful utilization in these areas. In Part 4, we provide an outlook on the future trends in lignin-based hydrogel research and their biomedical applications. We examine ongoing efforts to improve lignin extraction and isolation techniques, as well as the development of novel lignin-based hydrogel formulations. Additionally, we reflect on the potential impact of lignin-based materials, emphasizing their importance in advancing sustainable and innovative solutions in diverse fields. Finally, the Conclusions section summarizes the key findings and implications discussed in this review paper. We highlight the significant contributions of lignin biopolymers and lignin-based hydrogels in various applications.

## 2. Lignin Biopolymer

Lignin is one of the three major components of the cell wall of lignocellulosic biomaterials [35]. Its name was introduced as early as 1819 by Candolle (1778–1841), derived from the Latin lignum, meaning wood [39]. It can be isolated from various lignocellulosic materials, including agricultural residues, energy crops, wood residues, etc. [40,41]. The molecular mass of isolated lignin ranges from 1000 to 20,000 g/mol [42]. Lignin serves as a structural material that adds strength, rigidity, and impermeability to plants’ cell walls and facilitates the transport of water and solutes through the vascular system [43]. Additionally, depending on the functional group contents, it exhibits antioxidant and antimicrobial properties, thermal stability, an anti-inflammatory effect, biocompatibility, and low cytotoxicity [21,22]. Furthermore, it plays a vital role in protecting plants against biochemical stresses by inhibiting the enzymatic degradation of other components and providing a physical and chemical barrier that protects the plant tissue from terrestrial animals and microorganisms [22]. Therefore, lignin is both biologically and mechanically suitable for the preparation of numerous biomaterials, such as hydrogels [44]. Despite this, the precise structure of lignin is still unknown [22].

### 2.1. Processing Methods for Lignin Extraction and Isolation

Lignin was extracted for the first time by Bjökman in 1956 using a dioxane–water mixture [39]. Generally, different ways of extracting lignin from different biomass resources have been defined [45]. Several processes have been developed in the past to isolate lignin from other lignocellulosic biomass components, yielding different types of lignin. Reaction time, temperature, solvent concentration, and the type of raw materials are some factors that affect the extraction yield and the physicochemical properties of lignin [46]. The lignocellulosic biomass can undergo a preliminary step like acid hydrolysis [47], mechanical, or hot-water pretreatments [48,49], among others, in order to solubilize the hemicellulose fraction (Figure 1).

Lignin can be isolated in various forms by different extraction processes and can be classified into sulfur-containing and sulfur-free technical lignins [50]. Typically, extracting raw lignin from lignocellulosic biomass results in lignin’s fragmentation into numerous mixtures of erratic components [47,51]. Moreover, several pretreatment methods, classified into chemical, physicochemical, and enzymatic pretreatments, have been developed to investigate and allow the isolation and recovery of lignin from lignocellulosic biomass [52]. Lignin can be obtained using kraft, sulfite, alkaline, steam explosion, or hydrolysis processes [53]. In addition, lignin biopolymers can be extracted using “greener solvents”, such as ionic liquids and deep eutectic solvents [54]. After extraction, several post-treatments can be performed to improve the purity of the obtained lignins [45].

According to the literature, numerous methods for extracting wood lignin have been developed, but they have not yet been used industrially [55]. Among these, four main organosolv pulping techniques (i.e., the Alcell^®^, ASAM, Organocell, and Acetosolv processes) have been shown to yield highly pure lignin mixtures with excellent performance characteristics such as structural optimization, low inorganic impurities, and low molecular weight, thereby opening up new potential applications [45,56]:Alcell^®^ process: ethanol and solvent pulping;ASAM process: alkaline sulfite anthraquinone methanol pulping;Organocell process: methanol pulping followed by sodium hydroxide and anthraquinone pulping;Acetosolv process: acetic acid, hydrochloric acid, or formic acid pulping.

### 2.2. Composition and Structure

Lignin is an amorphous, three-dimensional (3D), highly crosslinked aromatic biopolymer synthesized mainly from three primary monolignols called p-coumaryl alcohol, coniferyl alcohol, and sinapyl alcohol (Figure 2), which generate the different types of lignin subunits by enzymatic polymerization, namely, p-hydroxyphenyl (H), guaiacyl (G), and syringyl (S), respectively [57].

These subunits contain various chemical groups, such as hydroxyl, carboxyl, carbonyl, and methoxy groups, which are active sites for further chemical modification and lignin utilization [35]. They differ in the number of methoxy groups [44]. In addition, lignin is covalently linked to cellulose and hemicellulose by phenyl glycoside, benzyl ether, and benzyl ester bonds to form lignin–carbohydrate complexes (LCCs) [36,59]. The nature and composition of lignin vary according to the plant species, environmental conditions of growth, seasonal conditions for its harvest, and the extraction process used to separate it from lignin–carbohydrate complexes [1,46,60]. Depending on the type of plant species, other monolignols may be present at infinitesimal concentrations, forming additional inter-unit linkages that make it difficult to estimate the degree of polymerization [42,61]. Thus, owing to its compositional diversity, the macromolecular structure of lignin remains elusive.

These building blocks are connected by different types of linkages, mainly β-O-4 ether linkages, which account for more than 50% of lignin’s linkage structure and are a crucial target for most degradation mechanisms. Other bonds include β-5 phenylcoumaran, β-β resinol, α-O-4 ether, 4-O-5 diphenyl ether, 5-5 biphenyl, and β-1 diphenyl methane, which make up smaller percentages (Figure 3) [62].

### 2.3. Lignin Properties for Biomedical Applications

Numerous studies have highlighted lignin as a renewable, biodegradable, biocompatible, and safe biopolymer that can be applied as an antioxidant, antimicrobial, anti-ultraviolet agent, and hemostatic agent, which can be attributed to its functional groups, i.e., aromatic rings; aliphatic and phenolic hydroxyl, carboxyl, carbonyl, and methoxy groups [63]. This may open up new perspectives in the formulation of various materials for pharmaceutical and biomedical applications [22,43,64].

#### 2.3.1. Antimicrobial Activity

Various investigations have suggested that lignin could be a promising green replacement for the fossil-based agents that are useful against dangerous microorganisms [65]. It was reported that the methoxy and epoxy groups were responsible for the antibacterial activity of lignin, which is attained by the contact of these compounds with bacteria, which leads to damage to the cell membrane and the lysis of the bacteria [37,66]. This raised the possibility of attempts to develop high-value antibacterial lignin-based biomaterials such as hydrogels, films, nanofibers, and nanoparticles. However, the antibacterial potential of lignin depends on the preparation methods employed [67,68]. For instance, lignin nanoparticles incorporated into polylactic acid (PLA) revealed an innovative capacity to inhibit bacterial growth over time [69]. Additionally, in order to prevent bacterial adhesion, several lignin-based composites have been developed, such as a lignin-based hydrogel supplemented with silver nanoparticles that showed inhibitory effects on both *Staphylococcus aureus* (*S. aureus*) and *Escherichia coli* (*E. coli*) [70,71]. Another study reported that silica–lignin hybrid materials modified with nanosilver effectively inhibited the growth of *Pseudomonas aeruginosa* (*P. aeruginosa*), a dangerous human pathogen [72]. Additionally, lignin displayed potent antiviral activity [38]. For example, Qiu et al. found that lignosulfonic acid, obtained through the sulfite delignification process, showed antiviral activity against human immunodeficiency virus (HIV) and herpes simplex virus (HSV) [73].

#### 2.3.2. Antioxidant Activity

The authors concluded that the antioxidant activity of lignin is positively correlated with the number of phenolic hydroxyl groups and methoxy groups [37,65]. These functional groups lead to the termination of the oxidative propagation reaction via hydrogen donation [62]. The antioxidant capacity of lignin has been exploited in medical, pharmaceutical (e.g., anticarcinogenic agent), and polymeric applications (e.g., thermal behavior enhancement) [74]. The high antioxidant potential of lignin has already been mentioned in previous works, making it a good alternative to replace cytotoxic synthetic antioxidants like butylated hydroxytoluene (BHT) or butylated hydroxyanisole (BHA), which are widely used in industries such as pharmaceuticals, cosmetics, and food [75]. Lignin’s antioxidant activity depends on its biomass source, molecular weight, polydispersity, extraction method, and post-treatment reactions, among other things [66,76,77]. Lignin’s antioxidant activity was evaluated by 2,2-diphenyl-1-picrylhydrazyl (DPPH) [78]. DPPH is a stable free radical that can be used to measure the radical-scavenging activity of antioxidants. Several researchers have highlighted the antioxidant potential of lignin; for example, Toh et al. reported that the autoxidation of linoleic acid was decreased by 50% in the presence of tea leaf lignin [79]. Kaur et al. reported that unmodified lignin from sugarcane bagasse had higher antioxidant activity than lignin that was chemically modified via acetylation and epoxidation [80]. However, due to lignin’s structural variability, variation in chemical composition, and the absence of clear guidance for structure–activity relationships, its commercial applications as an antioxidant agent are limited. Therefore, it was proposed to separate lignin biopolymer into fractions with low polydispersity and well-defined properties (i.e., molecular mass, polarity, number of phenolic hydroxyl groups, and other functionalities). Despite the success of fractionation in improving antioxidant activity, lignin-derived antioxidants still cannot compete with commercial phenolic antioxidants [81]. Due to the antioxidant and antimicrobial potential of lignin, it can be used for a broad variety of potential applications, such as drug delivery in cancer therapy [37,69].

#### 2.3.3. Anti-Ultraviolet Capacity

Lignin is a green and ideal material that exhibits high ultraviolet-absorbent properties because of its excellent oxidation resistance [82]. These are attributed to the functional groups of lignin’s backbone, including chromophore functional groups such as quinones, phenolics, ketones, and conjugated double bonds [66,83]. Wang et al. described the highest ultraviolet (UV) absorption performance of soda lignin from palm fiber. Their research revealed that the branched aromatic structure of lignin contributed to the superior UV-blocking performance of palm fiber [84]. These properties can be exploited in the preparation of different products as an alternative to conventional petroleum-based materials, such as films for food packaging and biomedical materials [22]. Sadeghifar et al. demonstrated the significant role of lignin as a biopolymer that contains UV-absorbing functional groups to produce a renewable-based cellulose–lignin UV-light-blocking film that was stable against elevated temperatures and UV irradiation [85]. Additionally, lignin biopolymer can be added to commercial sunscreen products with the purpose of increasing the sun protection factor (SPF) [57]. Qian et al. prepared different sizes of lignin colloidal spheres via the self-assembly method and mixed them with pure skin creams in order to develop lignin-based sunscreens. The results indicated that the sunscreen performance of creams with colloidal spheres was enhanced compared with those blended with original lignin [86].

#### 2.3.4. Other Properties

Lignin exhibits several other critical properties that make it attractive for medical applications, including anticoagulation, cholesterol reduction, and anti-hyperglycemic effects.

With the goal of developing better molecules as regulators of coagulation, several researchers have suggested that low-molecular-weight lignins (LMWLs), in polydisperse and heterogeneous preparations, mimic the polyanionic scaffold of low-molecular-weight heparins (LMWHs) [38,87,88,89]. LMWLs exhibit an anticoagulation profile in human plasma and blood that is similar to that of LMWHs [88].

Moreover, lignins have shown potential biological activities, such as the capability of reducing cholesterol by binding to bile acids in the intestine [90]. Additionally, modified alkali lignin showed significant in vitro α-amylase-inhibitory activity, thereby indicating that it potentially has anti-hyperglycemic properties and can inhibit the evolution of diabetic disease [38].

The critical properties of lignin biopolymer show its versatility and promise to promote healthcare. Although these features are encouraging, it is crucial to keep in mind that more investigations are required to completely comprehend the mechanisms underlying these outcomes, and to improve lignin derivatives for medical applications.

## 3. Lignin-Based Hydrogels

By benefiting from the different attractive advantages and sustainable properties of lignin biopolymer (e.g., antioxidant and antimicrobial properties, anti-inflammatory effect, biocompatibility, and low cytotoxicity [21,22]), it can serve as an excellent building block for fabricating hydrogels [91] that can be widely used in the biomedical field.

### 3.1. Hydrogels

Hydrogels are commonly known as crosslinked hydrophilic polymeric materials in the form of a 3D viscoelastic or elastic network created from natural or synthetic macromolecules [1]. The first hydrogel was synthesized by Grindlay and Clagett in 1949 [92]. They used a hydrogel made from poly(vinyl alcohol) (PVA) and formaldehyde as a plastic sponge prosthesis for use after pneumonectomy [92]. Moreover, 11 years later, a poly-2-hydroxyethylmethacrylate (PHEMA) hydrogel marked a significant turning point in the production of hydrogel materials [18]. Wichterle and Lim proposed this hydrogel as a synthetic biocompatible material useful for contact lens applications [18].

Hydrogels have attracted the significant interest of many researchers in a wide variety of applications, typically in four major sectors: biomedical [20,93], agriculture [94], environment [95], and electronics [41]. Biomedical applications include drug delivery, 3D cultures, tissue implants, tissue regeneration, contact lenses, and uses in healthcare products due to their unique properties such as softness, flexibility, biocompatibility, and the affinity to absorb water, organic solvents, and biological fluids up to one thousand times their own dry weight without being dissolved, due to the presence of chemical and physical crosslinking in their structure [18,91,96]. They stay in balance in an aqueous environment due to the balance between the elastic forces of their 3D interconnected structure and the osmotic forces of the surrounding liquid (Figure 4) [20]. Other interesting properties of these materials include their ability to control the diffusion process and to perform dramatic volume transitions in response to specific environmental stimuli, which can be physical, chemical, or biological [97,98].

Hydrogels restore their original state when these environmental stimuli disappear [99]. They can be characterized by numerous physical parameters, such as size, elastic modulus, swelling, and degradation rate [96]. These materials can be classified according to their origin, preparation method, crosslinking nature, swelling capacity, and sources. However, classification based on the nature of crosslinking is the major factor that determines whether a hydrogel is chemically or physically crosslinked [98].

Chemically crosslinked networks involve covalent bonding that is permanent at junctions; this gives the final product more resistant mechanical properties. This type of hydrogel requires a crosslinking agent and free-radical polymerization in the presence of initiators [41]. Meanwhile, physical networks have reversible junctions that arise from either polymer chain entanglements or physical interactions such as ionic interactions, hydrogen bonds, hydrophobic forces, or Van Der Walls interactions, without any crosslinking agent [1,41]. Depending on the polymer source, hydrogels can be synthetic, natural, or hybrid [100]. Natural hydrogels, using biopolymers as building blocks, have beneficial properties that are favored by researchers in terms of biocompatibility, cost-effectiveness, and biodegradability. This is the case with lignin-based hydrogels, which are considered to be a more sustainable and environmentally friendly alternative to synthetic hydrogels [91] and are incredibly appreciated throughout biomedicine [101].

**Figure 4 ijms-24-13493-f004:**
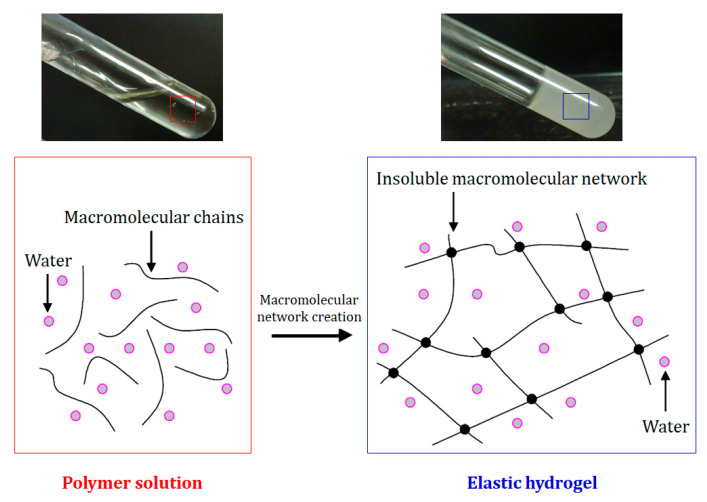
A schematic illustration of the principle of hydrogel formation based on hydrogel crosslinking, which forms an insoluble macromolecular network in the environmental fluid: The macromolecular network’s creation could be based on physical or chemical crosslinking (reprinted from Fatimi, 2021 [102]; Copyright© 2021 MDPI under the terms of the Creative Commons Attribution 4.0 International License).

### 3.2. Preparation of Lignin-Based Hydrogels

Lignin is an attractive raw material for hydrogel preparation, with significant sustainability and green chemical connotations [103]. Using lignin biopolymer alone to form a hydrogel cannot ensure good water retention and high mechanical properties; therefore, it is necessary to resort to copolymerization in order to obtain a better performance in these properties [30]. Lignin-based hydrogels can be synthesized by a variety of methods (e.g., interpenetrating polymer network of lignin and crosslinking monomer, (Appendix A), crosslinking grafted lignin and monomers (Appendix A), atom transfer radical polymerization (Appendix A), reversible addition–fragmentation transfer polymerization (Appendix A), etc.). It is significant to note that all of these methods for the synthesis of lignin-based hydrogels were emphasized by Meng et al. in 2019 [35].

These methods can be divided into two major groups: lignin-based hydrogels formed by chemical interactions, and lignin-based hydrogels formed by physical interactions. More precisely, chemically crosslinked hydrogels can be obtained by grafting, free-radical polymerization, click chemistry, enzymatic reactions, thermo-gelation, and radiation crosslinking. On the other hand, to create physically crosslinked hydrogels, changes in intermolecular interactions—such as ionic crosslinking, hydrophobic interactions, and hydrogen bonding—have been employed. Both chemical and physical crosslinking can be used to form hydrogels [96]. These processes can be initiated by temperature, ionic, gamma, or redox pairs [1].

Table 1 summarizes a review of relevant lignin-based hydrogels by chemical and physical interactions as a function of lignin’s roles (i.e., crosslinked unit or crosslinking agent).

#### 3.2.1. Lignin-Based Hydrogels by Chemical Interactions

Due to the chemical reactivity of lignin, most lignin-based hydrogels described in the literature were synthesized by chemical crosslinking [116]. The crosslinking of grafted lignin and monomers is a way to synthesize hydrogels, in which lignin and unsaturated monomers or functional compounds are grafted onto the main chain by the esterification reaction of lignin’s phenolic groups. The grafted lignin is then crosslinked with other unsaturated monomers, such as hydroxyethyl acrylate, to form hydrogels with good water retention [91]. Ma et al. used this approach to synthesize lignin-based hydrogels that could be used to remove lead ions from the human body because of their toxicity and their potential effects on the nervous, immune, renal, reproductive, and cardiovascular systems [117]. Other hydrogels were prepared by copolymerization of acrylic acid and other functional compounds such as organo-montmorillonite (OMt) with grafted lignin, which was synthesized by adding lignin in the presence of sodium hydroxide, with N,N’-ethylenebisacrylamide (NEBA) and ammonium persulfate (APS) as initiators [91]. Moreover, a novel lignin-based thermosensitive gel was prepared by thermal polymerization of phenolated alkali lignin and N-isopropyl acrylamide. These materials can be used to control pests and diseases. They can also release and recover chemicals through temperature changes in nature. In addition, lignin has the function of absorbing UV rays; this property has a good preservation effect for UV-decomposable drugs [118].

Sathawong et al. [107] recovered lignin from kraft black liquor by the acidic precipitation method and synthesized a lignin–agarose hydrogel with epichlorohydrin (ECH) as a crosslinking agent. Agarose is a natural polysaccharide extracted from the cell walls of certain *Rhodophyceae* algae. Even at low concentrations, it enhances system stability, making it suitable for hydrogel scaffolds, cartilage, neural tissue, and wound healing. Kraft lignin was also mixed with solid phenol and an aqueous solution of formaldehyde to form highly porous hydrogels [32].

Teng et al. [104] prepared a new lignin-based hydrogel from lignin amine and poly(ethylene glycol) diglycidylether (PEGDGE). Lignin amine was prepared via the Mannich reaction by grafting the amino groups onto sodium lignin sulfonate. These hydrogels showed superior swelling capacity, viscoelasticity, and shear properties [104]. The aminated lignin was also expected to react with poly(vinyl alcohol) (PVA) to form a hydrogel, in which silver nanoparticles were reduced in situ to enhance the antimicrobial properties of lignin against *S. aureus* and *E. coli* [71].

Lignosulfonate was used by Wu et al. [109] to prepare a superabsorbent hydrogel through graft copolymerization of magnesium lignosulfonate with acrylic acid and acrylamide, using potassium persulfate as an initiator and *N*,*N’*-methylenebisacrylamide (NMBA) as a crosslinker.

PVA is a hydrophilic, semicrystalline synthetic polymer that exhibits attractive properties such as a high degree of swelling, a rubbery or elastic nature, non-toxicity, and bioadhesivity, which make it a good candidate for the synthesis of hydrogels for biomedical applications. PVA was mixed with lignin or lignin–epoxy resins and ECH crosslinking agents to form hydrogels by covalent crosslinking [8]. Lignin–PVA hydrogels were also prepared by Wu et al. [110]. The lignin macromolecule was incorporated into the PVA hydrogels in the presence of ECH as a crosslinker to form a doubly crosslinked network exhibiting swelling ratios of up to 456 g/g, excellent water retention, and biodegradability. Wu et al. then proposed a structural mechanism for the lignin–PVA hydrogel (Figure 5). In this case, the hydroxyl group of PVA or lignin reacted with the epoxy group of ECH to form an ether bond, while hydrochloric acid was taken out of the other end of ECH to generate a new epoxy group. The previously mentioned reaction was carried out until the crosslinking process was finished, at which point the extra crosslinking agent was eventually changed into glycerol [119]. Additionally, the hydroxyl groups of lignin and PVA generated potent intermolecular hydrogen bonds [120], and PVA was able to crosslink with ECH to form a compact network structure. Finally, through ECH, lignin–PVA and PVA–PVA might crosslink with one another to form the lignin–PVA hydrogel [110].

Furthermore, Akhramez et al. confirmed in a recent study that, through ECH, lignin–PVA and PVA–PVA were crosslinked with one another to form the lignin–PVA hydrogel [111]. In this recent study, novel lignin-based hydrogels were developed for future applications in biomedical engineering using modified bagasse-sourced lignin [111]. Alkaline delignification was first used to recover lignin from sugarcane bagasse. The lignin fractions were then modified by salinization and acetylation reactions. The results demonstrated that modified lignin can be used to formulate hydrogels using PVA and ECH as the matrix and crosslinking agents, respectively. In addition, the salinization of lignin has made it possible to obtain hydrogels that are more efficient in terms of rheological properties and swelling compared to other formulated hydrogels [111].

Due to the antimicrobial activity of the lignin biopolymer, it serves as a potential drug-eluting antimicrobial coating for medical purposes. Lignin was combined with poly(ethylene glycol) (PEG) and poly(methyl vinyl ether-co-maleic acid) (PMVE/MA) through esterification reactions to form hydrogels, which demonstrated logarithmic reductions in the adherence of *S. aureus* and *Proteus mirabilis* (*P. mirabilis*) [113].

Lignin was combined with xanthan, which is a polysaccharide with excellent antioxidant and rheological properties, in the presence of ECH as a crosslinking agent in an alkaline medium to form xanthan–lignin hydrogels. These hydrogels showed promising applications in drug delivery [108]. It has been shown that the lignin type affects the final lignin–xanthan hydrogel’s morphology, which can be smooth or fibrillar depending on the hydrogel’s source. As an example, a fibrillar morphology was confirmed in the case of a hydrogel containing aspen wood lignin, which has the largest concentration of carboxylic acid groups and the lowest concentration of phenolic hydroxyl groups [108].

A superabsorbent cellulose–lignin hydrogel was prepared by Ciolacu et al. [33] by dissolving cellulose in an alkaline solution and further mixing it with lignin, followed by chemical crosslinking with ECH. As already mentioned above [110,119,120], Ciolacu et al. confirmed that the epoxy cycle from ECH opens in an alkaline environment and then bonds to the hydroxyl groups of cellulose or lignin. Following the displacement of the chloride, additional epoxy groups may emerge. These groups can then react with the hydroxyl groups on adjacent cellulose or lignin chains to generate crosslinked cellulose and lignin. These hydrogels were then evaluated for the controlled release of polyphenols. The results demonstrated the superabsorbent character of lignin hydrogels [33].

To prove the water-absorbing capacities and the superabsorbent character of lignin hydrogels, other studies have been conducted on lignin-based hydrogels obtained by chemical interactions. In this way, a superadsorbent hydrogel was prepared from lignin and montmorillonite (Mt) for copper(II) ion removal. Lignin and Mt were mixed in the presence of a redox initiator with acrylic acid and the crosslinking agent NMBA [112]. Furthermore, high water-absorbing capacities were measured for lignin hydrogels prepared by crosslinking PMVE/MA and different technical lignins in ammonium and sodium hydroxide solutions. The fundamental mechanism lies in the esterification reaction of the hydroxyl groups of lignin with the carboxylic acids of maleic acid units [103]. However, in this study, lignin biopolymer was used as a crosslinking agent.

Several methods for the preparation of crosslinked hydrogels are based on free-radical reactions, in which lignin hydroxyl groups form radicals in the presence of initiators that further form grafting structures with monomers or polymer chains. The obtained product penetrated into the network that was formed from monomers to synthesize interpenetrating polymer network hydrogels [1]. Based on this strategy, El-Zawawy et al. [105] prepared lignin hydrogels in the absence of an external crosslinker by grafting alkaline or kraft lignin to acrylamide and PVA to form AM-PVA-g-lignin copolymers and then mixing them with an acrylamide monomer. More recently, the same authors [106] repeated these formulations by adding NMBA as a crosslinker to achieve extra crosslinking. In another study, polyacrylamide (PAAm) was also used with lignin to enhance its mechanical properties [121]. Ultrasonic treatment for lignin nanoparticle dispersion has been realized, and in situ free-radical polymerization for a lignin–PAAm hydrogel has been carried out.

Sodium-lignosulfonate-grafted poly(acrylic acid-co-poly (vinyl pyrrolidone)) (PAA/PVP) is a smart hydrogel that was prepared by radical polymerization with the ultrasonic assistance of acrylic acid in the presence of acid-activated lignosulfonate and poly(vinyl pyrrolidone) (PVP). The swelling behavior and pH sensitivity of the developed hydrogel were investigated. This hydrogel showed a high water-swelling property, good pH responsiveness, and excellent sustained-release performance in the intestine [25].

#### 3.2.2. Lignin-Based Hydrogels by Physical Interactions

In order to avoid the use of toxic chemical reagents, physical crosslinking methods are used for lignin hydrogel preparation as a greener and more economical approach [122].

Oveissi et al. [24] used lignin for additional crosslinking of hydrophilic polyether-based polyurethane (HPU) hydrogels in order to improve their mechanical strength and processability by forming hydrogen bonds between the PU and the polar sites of lignin’s backbone. The authors confirmed that hydrogen bonding was the dominant toughening mechanism, and they elucidated the toughening mechanism by applying the Lake–Thomas and sequential deboning theories. Finally, these hydrogels have demonstrated good biocompatibility with primary human dermal fibroblasts and have been proposed for scalable fabrication methods such as 3D printing, fiber spinning, and film casting [24].

Another preparation method for lignin-based hydrogels was successfully introduced by Ravishanker et al. [23], who prepared a physical hydrogel with greater viscoelastic properties by mixing an aqueous–acidic solution of chitosan with alkali lignin, making it highly desirable for application as scaffolds in tissue engineering and wound healing. The incorporation of lignin could potentially improve the shear strength and viscosity of chitosan [23].

Lignin was also used as a physical crosslinking agent for the preparation of a lignin–chitosan–PVA composite hydrogel [115]. In this study, sulfonate groups in the chemical structure of lignin formed ionic bonds with the amino groups of chitosan, thereby increasing the mechanical strength of the hydrogel. The developed hydrogel showed excellent biocompatibility, antimicrobial activity, and high mechanical strength. Zhang et al. confirmed that lignin–chitosan–PVA hydrogels exhibit potential for a wide range of medical applications, such as packaging expensive drugs [115]. Figure 6 shows the crosslinking mechanism of lignin–chitosan–PVA hydrogel, in which active groups of lignin such as hydroxyl, carboxyl, and sulfonic groups can form hydrogen bonds (formed by the hydroxyl of PVA and the hydroxyl of lignin) and ionic bonds (formed by the amino group of chitosan and the sulfonic group of lignin) [115].

On the other hand, Huang et al. [114] proposed novel crosslinked hydrogels based on PVA and hydroxyethylcellulose (HEC) in the presence of lignin as a plasticizer and borax as a crosslinker. Generally, PVA exhibits a hydrogen-bonding character that makes it able to form PVA–borax hydrogels with a physically crosslinked network via the hydrogen bond and a reversible didiol–borax complex [123]. However, due to their weak mechanical strength, typical PVA–borax hydrogels have a limited range of uses. To overcome these issues, Huang et al. proposed the introduction of lignin, since HEC and lignin with rich hydroxyl groups have good compatibility with PVA. Thanks to the interaction of lignin molecules and flexible PVA chains or HEC chains via hydrogen bonds and physical junctions in the presence of borax, the results effectively indicated that the lignin-based composite hydrogels exhibited elastic-like behavior, implying a rather stable and strong molecular network, as well as a reversible thermosensitive property due to thermal-induced water evaporation and reversible and exothermic reactions. These hydrogels possessed great stretchability, thermosensitivity, electrical conductivity, and self-healing capability, which made them more suitable for applications in the fields of 3D printing and wearable electronic devices [114].

Excellent mechanical strength is a major advantage of chemically crosslinked hydrogels; however, as mentioned above, the formulation of these hydrogels may require toxic and expensive crosslinking agents. Despite their weaknesses, physical hydrogels have attracted much interest in recent years due to their environmentally friendly and economical synthesis methods. Currently, several research groups are looking for new chemistries for the preparation of lignin-based hydrogels to improve their performance. For instance, Larrañeta et al. [113] prepared hydrogels using an eco-friendly method by combining lignin with PMVE/MA, polyacids, and PEG through esterification reactions. These hydrogels have been used in drug delivery applications.

## 4. Biomedical Applications of Lignin-Based Hydrogels

Lignin has been proven to be an attractive candidate as a backbone polymer in the development of hydrogels for different biomedical applications [8]. The main areas of lignin-based hydrogel applications are tissue engineering, wound dressing, drug delivery systems, and 3D bioprinting [27,28].

Table 2 summarizes some biomedical applications of lignin-based hydrogels in the areas of tissue engineering, wound healing, drug delivery, and 3D bioprinting.

### 4.1. Tissue Engineering

Tissue engineering is an interdisciplinary area that combines the use of cells, growth factors, and innovative scaffolds to recover, replace, improve, or preserve a specific tissue or organ that is functioning suboptimally as a result of either chronic disease or acute trauma [66,101,129,130]. The ideal scaffolds should require potential biocompatibility, biodegradability, an interconnected porous structure, and excellent rheological properties [131]. Lignin-based hydrogels are among the optional scaffolds used for tissue regeneration, owing to their bioactivity and their 3D structure mimicking the native extracellular matrix [12,131,132].

Furthermore, these biomaterials provide an adequate porous microenvironment permitting gas diffusion and metabolite exchange in order to support the proliferation and differentiation of encapsulated cells, thereby developing the damaged tissue without open surgery [133]. For example, hydrogels of chitosan and alkali lignin show attractive properties such as biocompatibility and a conductive surface for cell attachment and growth, making them highly desirable for applications in tissue engineering [12]. In this way, Ravishankar et al. proposed a novel formulation of a biocompatible hydrogel using chitosan and alkali lignin [23]. Figure 7 summarizes the results of cell viability of studied mesenchymal stem cells, a morphological image indicating cell adhesion on chitosan–alkali lignin hydrogel, as well as fluorescein diacetate (FDA)- and 4′,6-diamidino-2-phenylindole (DAPI)-stained fluorescence images of cell adhesion on lignin-based hydrogel. In conclusion, the prepared chitosan–alkali lignin hydrogel in vitro was found to be benign to mesenchymal stem cells, and it was non-toxic to zebrafish in vivo at a concentration of 10 mg/mL [23].

For applications in tissue engineering, another lignin-based hydrogel has demonstrated excellent mechanical properties and non-toxicity [121]. Chen et al. developed different lignin–PAAm hydrogels by using an ultrasonic treatment for lignin nanoparticle dispersion, followed by in situ free-radical polymerization. The lignin-based hydrogels were then co-cultured with human esophageal squamous carcinoma cells, and a live/dead viability assay used to study the viability of the cells in the hydrogels confirmed that the cells retained a high level of cell viability and preserved their ability to proliferate [121].

An alginate biopolymer was associated with lignin to form a lignin-based foamed hydrogel with good textural and mechanical properties, as well as high water uptake [124]. The key idea of the approach proposed by the authors is the exposure of alginate and lignin in an aqueous–alkali solution containing calcium carbonate to carbon dioxide. Since gelation was based on carbon dioxide foaming, the obtained dual porosity of the hydrogel was advantageous for good cell adhesion properties, without compromising cell viability. Owing to its non-cytotoxicity and good cell adhesion, this developed lignin-based hydrogel is an attractive candidate for a wide range of applications, including tissue engineering and regenerative medicine [124].

### 4.2. Wound Healing

Wound healing is a natural physiological process related to structural damage to tissue and is generally divided into four stages: hemostasis, inflammation, proliferation, and remodeling, in which cells and endogenous factors are involved. In order to address these problems, with the aim of accelerating and performing wound repair, several materials have been developed as wound dressings [134,135]. Among them, hydrogel biomaterials have great advantages in the treatment of different types of wounds, such as incisional wounds and excisional wounds [136]. Due to their porous structure, soft nature, flexibility, stretchability, biocompatibility, and high water content, hydrogels can absorb a large amount of exudate, maintain a moist environment on the wound surface, and act as carriers for bioactive substances and cells, which are expected to prevent infection and desiccation, alleviate pain, activate immune cells to accelerate the healing process, and avoid scar formation [129,137,138]. 

An ideal wound dressing should have the ability to maintain an appropriate temperature to promote blood flow to the wound bed; it must be sterile, non-toxic, and non-allergenic [139]. In the past few years, hydrogels have passed from their classical use as primary wound dressings to effective regenerative templates capable of promoting wound closure and/or skin regeneration following burn injuries and other skin lesions [140]. The wound-healing performance is enhanced through the integration of antioxidant and antibacterial agents in the formulation of hydrogel dressings; this is the case with lignin biopolymers [137,140]. Compared with traditional passive wound dressings like gauze, bondage, and cotton, which cannot provide a moist healing environment and will cause reinjury when they are removed, lignin-based hydrogels can remove undesired substances, resist microbial infections, and provide wound protection with a moist environment around the wound, allowing for gas exchanges and absorbing exudate. They can also be removed without causing secondary physical damage to the wound [137,141]. For example, lignin–chitosan–PVA hydrogels served as good wound dressings in terms of mechanical strength, protein adsorption, and cell proliferation [115]. Additionally, injectable hydrogels have good fluidity and biocompatibility, as well as excellent permeability for metabolites, which makes them more suitable for wound repair and reduces the need for invasive surgery. They form a gel in situ that can fill the wound in three dimensions. This enables them to reach deep and irregular wounds that traditional hydrogels cannot fill [141,142]. 

Six years ago, Mahata et al. proposed a biocompatible lignin-based hydrogel as an ointment for anti-infective activity [125]. The lignin-grafted polyoxazoline-conjugated triazole was obtained by polymerization and covalent modification. The resulting hydrogel was capable of supporting cellular anti-inflammatory activity. The study confirmed that the novel lignin-based hydrogel demonstrated the ability to prevent infection of burn wounds, aid healing, and act as an anti-inflammatory dressing material in vivo [125].

In 2018, Oveissi et al. used lignin for additional crosslinking of HPU hydrogels in order to improve their mechanical strength and processing ability by forming hydrogen bonds between the PU and the polar sites of lignin’s backbone [24]. Figure 8 presents the application of the developed hydrogel for wound healing. The authors proposed the lignin–HPU hydrogel as a film and patch on the arm. Furthermore, the hydrogel film peeled off without any attached hair and without any pain. Additionally, the good viability of human dermal fibroblast cells on the lignin–HPU film demonstrated good biocompatibility compared to controls over three days [24].

In 2019, a chitosan–PVA composite hydrogel was modified to satisfy the requirements of wound dressing and as an environmental conditioner to accelerate wound healing [115]. Zhang et al. added lignin to this composite hydrogel as a crosslinking agent. Based on the results of this study, the introduction of lignin as a crosslinking agent effectively improved the mechanical strength, protein adsorption capacity, and wound environmental regulation ability of the chitosan–PVA hydrogel [115]. The in vivo study also confirmed that this lignin-based hydrogel could be used for highly efficient skin wound care and management. Effectively, data analysis of the murine wound model demonstrated that the lignin–chitosan–PVA hydrogel could maintain a moist healing environment and enable faster healing than the chitosan–PVA hydrogel (Figure 9).

### 4.3. Drug Delivery

Hydrogels are among the most widely investigated systems for successful drug delivery [132,143]. The drug release mechanisms of hydrogels depend on several factors, which can be related to solute characteristics, formulation compositions, or polymer properties [144]. Due to the biocompatibility, antioxidant and antimicrobial activity, low toxicity, chemical composition, and stimulus response of lignin-based hydrogels, numerous studies have described their potential as drug delivery systems [145]. This technology aims to achieve higher therapeutic efficiency of a natural compound in an exact diseased location in a controlled manner, without causing severe toxicological effects [68]. The rate of drug release depends on the hydrogel network’s density [146]. Their porous structure makes them highly attractive drug delivery vehicles; thus, they enhance the therapeutic outcome of drug delivery and have found enormous clinical use [147]. Hydrogels have great potential for drug release in a controlled manner in different ways, depending on the release rate-limiting step: (i) swelling-controlled, which depends on the time necessary for the solvent to integrate the polymeric matrix and form the gel layer; (ii) diffusion-controlled, which depends on the dug diffusivity across the polymeric matrix; and (iii) chemically controlled, which depends on reactions occurring inside the polymeric matrix [144,148].

Various polysaccharide-based hydrogels, including lignin, have been created for use in drug delivery [149]. Using lignin, hemicellulose, and starch in a reactive extrusion method with citric acid and sodium hypophosphite as catalysts, Farhat et al. studied a variety of hydrogel formulations used as drug delivery systems for pharmaceutical drugs [149]. Previously synthesized lignin-based hydrogels showed promising drug delivery performance. For that, lignin was combined with PMVE/MA and PEG to form a highly swellable hydrogel for the controlled release of hydrophobic curcumin for up to 4 days. This hydrogel demonstrated logarithmic reductions in the adhesion of *S. aureus* and *P. mirabilis* [113]. Furthermore, Raschip et al. manufactured a lignin–xanthan hydrogel using ECH as a crosslinking agent for the controlled release of hydrophilic bisoprolol fumarate for high blood pressure and heart failure treatments [108]. In addition, lignin was mixed with cellulose and ECH to form hydrogels with high swelling capacities used for the release of polyphenols [68].

Another way to formulate lignin-based hydrogels for drug delivery systems concerns sodium-lignosulfonate-grafted poly(acrylic acid-co-poly(vinyl pyrrolidone)) (SLS-g-P(AA-co-PVP)) hydrogels [25]. This study is an example of how using lignin-based synthetic-drug-loaded hydrogels is one of the potential approaches to recycling and valorizing lignosulfonate, which is considered to be a waste product of pulping. In this way, Wang et al. investigated the swelling behavior and cumulative release rate as a function of the pH sensitivity of the SLS-g-P(AA-co-PVP) hydrogel [25]. Amoxicillin was used as a model drug to study the release properties in enzyme-free simulated gastrointestinal fluids (i.e., simulated gastric fluids (SGF) and simulated intestinal fluids (SIF)), and it exhibited favorable pH sensitivity and controllable release behavior in vitro. According to the results, the amoxicillin encapsulated with SLS-g-P(AA-co-PVP) hydrogel showed a better release effect in the SIF (pH = 7.4) than in the SGF (pH = 1.2). It can be seen that by adjusting the pH value of the medium, the release behavior could be controlled (Figure 10A). Due to the ionization of carboxylic groups in polymer chains at pH 7.4, the hydrogels swelled, and the hydrogen bonds between the amoxicillin and polymer chains broke down more often as the pH of the release medium increased. Figure 10B presents the results of the initial portion of the release curves. Compared to theoretical values, the authors confirmed an anomalous transport of amoxicillin. In other words, this concerns superposition controlled by both swelling and diffusion with case II transport, which is related to polymer relaxation during hydrogel swelling and Fickian-type diffusion, respectively. In addition to increasing the number of accessible binding sites (i.e., free hydroxyl groups), grafting sodium lignosulfonate also created a more porous 3D network of the hydrogel, which is helpful for loading more drug onto the hydrogel (Figure 10C,D) [25].

### 4.4. Three-Dimensional (3D) Bioprinting

Compared with traditional tissue engineering methods, 3D bioprinting technology shows promising advantages in tissue engineering and regenerative medicine [150]. The main difference is the use of bio-inks that are deposited layer by layer in a specific pattern that mimics native tissues and organs [20,133,151,152,153]. This emerging technique involves the creation of cell-laden structures through the layered deposition of bio-inks in vitro and in vivo [133]. It is a computer-assisted process that can control and plan the cellular and biomaterial geometries and special distribution in order to create anatomically correct biological structures [154,155]. An inherent characteristic of this technique is the ability to reliably and accurately place small volumes of materials and cells in specific locations repeatedly [155]. It is also possible to create personalized macroscopic and microscopic constructs at different scales that match patient anatomy [156]. More recently, Reynolds et al. introduced a new microporogen-structured matrix for embedded bioprinting that was able to tailor rheology, printing behavior, and porosity [153]. The integration of such a matrix opens up new opportunities for the in vitro biofabrication of complex tissue microenvironments that may be extensively used in tissue engineering for therapeutic purposes, disease modeling, and drug discovery [153].

The widely used 3D bioprinting technologies mainly include cellular inkjet printing, extrusion-based technologies, soft lithography, and laser-induced forward transfer [20,155,157,158]. Hydrogels have been intensively explored as bio-inks due to their promising properties, including biofunctionality, physicochemical properties, and rheological properties, which are required to provide structural support and prevent cell damage [20]. Bio-inks have been defined as a mixture of materials and biological molecules or cells suitable for processing by automated biofabrication technology [159]. Additionally, due to the non-cytotoxicity, biocompatibility, biodegradability, mechanical strength, and reactivity of the lignin biopolymer, it has been considered to be an excellent candidate to manufacture hydrogels for 3D bioprinting applications [160]. In this way, based on the literature, some significant studies have demonstrated that lignin-based hydrogels with improved printability and required performance as biomaterial inks have been used as valuable additives for 3D bioprinting applications [34].

Lignin-based hydrogels as biomaterial inks have been extruded from a 3D printer nozzle to print various patterns [24]. In this study, Oveissi et al. used lignin for additional crosslinking of hydrogels in order to improve their mechanical strength and processing ability by forming hydrogen bonds between the PU and the polar sites of lignin’s backbone [24]. After the lignin’s addition, the hydrogels showed an increase in Young’s modulus and fracture energy, as well as improved lap shear adhesiveness. Finally, direct ink writing has been realized with dry-spun hydrogel fibers, and various patterns have been printed (Figure 11).

Other interesting studies have been carried out to prove that lignin-based hydrogels could be used as suitable biomaterial inks [34]. Jiang et al. successfully printed scaffolds made up of alkali lignin crosslinked with Pluronic F127. The resultant printed structure exhibited stiffer and more water-resistant behavior [161]. Dominguez-Robles et al. used PLA–lignin to print wound dressings according to patients’ needs [26].

For cartilage tissue repair, a combination of gellan gum and lignin was tested as a bioprintable hydrogel utilizing the extrusion process [126]. It was revealed that the addition of lignin demonstrated good rheological properties in terms of shear-thinning behavior and printability (Figure 12). As a biological result, according to Bonifacio et al., the chondrogenic potential of the 3D structure was satisfactory compared to gellan gum hydrogels without lignin, which were used as controls [126].

Concerning 3D bioprinting of soft biological tissue, three years ago, a novel hydrogel-based bio-ink was made from lignin, cellulose, and alginate [127]. Cell viability assays were carried out on the bioprinted scaffold using a hepatocellular carcinoma cell line, and the results revealed no negative effect. Furthermore, results confirmed that the addition of lignin could help increase the viscosity and improve the printability and shape stability of this composite hydrogel. Zhang et al. finally confirmed that these scaffolds have high potential in soft-tissue engineering and regenerative medicine applications [127].

In the same year, a digital light processing technique for 3D bioprinting was used to print a lignin-based hydrogel (i.e., esterified dealkaline lignin with ethyl 4-(dimethylamino)benzoate as a co-photoinitiator). In this study, Zhang et al. esterified dealkaline lignin to improve its photoinitiation [128]. The 3D-printed structures were tested in vitro with fibroblasts, and live/dead staining demonstrated that the cells showed better proliferation [128].

## 5. Future Trends

The highly desirable properties of lignin-based hydrogels make them excellent candidates for biomedical applications. Our review shows their potential in the fields of wound healing, drug delivery, tissue engineering, and 3D bioprinting. For example, Ravishankar et al. [23] investigated the potential of lignin-based hydrogels prepared by mixing lignin with chitosan for tissue engineering and wound healing. On the other hand, Zhang et al. [128] mixed lignin with alginate and cellulose to form hydrogels for 3D printing. Additionally, Raschip et al. [108] prepared lignin–xanthan hydrogels for drug delivery using ECH as a crosslinker. Further important progress is required for developing new formulations or improving their performance in order for them to be more involved in the biomedical field in the future, especially in 3D bioprinting, as an innovative technology that has revealed an insightful impact on regenerative medicine.

Different techniques have been developed to overcome the recalcitrant nature of lignocellulosic biomass and extract lignin biopolymer. Lignin has gained considerable interest owing to its attractive properties. These properties may be more beneficial when including lignin in the preparation of highly desired value-added products, including hydrogels. In this respect, lignin-based hydrogels have gained significant attention in recent years due to their unique characteristics and potential properties. These hydrogels provide a tunable platform to overcome several limitations in the biomedical field in terms of healthcare improvement. Moreover, they offer several advantages, including biocompatibility, biodegradability, mechanical strength, etc., making them attractive for various biomedical applications.

Actually, the formulation of lignin-based hydrogels is always associated with other polymers and biopolymers (e.g., PVA, PAAm, PU, chitosan, alginate, xanthan, cellulose, agarose, gellan gum, etc.). In summary, PVA provides excellent biocompatibility, water retention ability, and film-forming properties. PAAm offers excellent water absorption properties and mechanical stability. PU offers excellent mechanical properties, flexibility, and biocompatibility. Chitosan provides antimicrobial properties, biocompatibility, and biodegradability. Alginate provides excellent gelation properties and biocompatibility. Xanthan provides viscosity and elasticity to the hydrogel matrix, improving its overall performance. The addition of cellulose enhances the water absorption capacity and mechanical strength of lignin-based hydrogels. Agarose provides excellent gelation properties, biocompatibility, and high water uptake ability. Gellan gum offers excellent gelation properties and controlled-release characteristics.

Future trends in lignin-based hydrogel research involve further optimization of their properties and functionalities through association with other polymers and biopolymers. Researchers are likely to focus on developing composite hydrogels that offer enhanced mechanical strength, stability, biocompatibility, and controlled-release properties. Moreover, the exploration of novel combinations and the incorporation of advanced techniques such as 3D printing and surface modification are expected to expand the applications of lignin-based hydrogels in tissue engineering, wound healing, drug delivery, and other biomedical fields. Continued research and development in this field are expected to expand the range of applications and drive future trends in lignin-based hydrogel research.

## 6. Conclusions

Lignin-based hydrogels show great promise for biomedical applications. By associating lignin with other polymers and biopolymers, researchers can tailor the properties of these hydrogels to suit specific applications. The combination of lignin with these polymers can enhance the mechanical strength, stability, gelation and viscoelasticity properties, film-forming properties, water uptake ability, and controlled-release characteristics of the hydrogels, among other properties, making them attractive for tissue engineering, wound healing, drug delivery systems, and 3D bioprinting. By providing a comprehensive review of the literature, this paper aims to contribute to the understanding of lignin biopolymers and their potential applications as hydrogels. This review paper provides a detailed overview of different types of lignin-based hydrogels and their preparation methods, as well as their applications in different biomedical fields. It could serve as a valuable resource for researchers and professionals interested in the utilization of lignin-based materials in various industries, promoting the development of sustainable and advanced technologies.

## Figures and Tables

**Figure 1 ijms-24-13493-f001:**
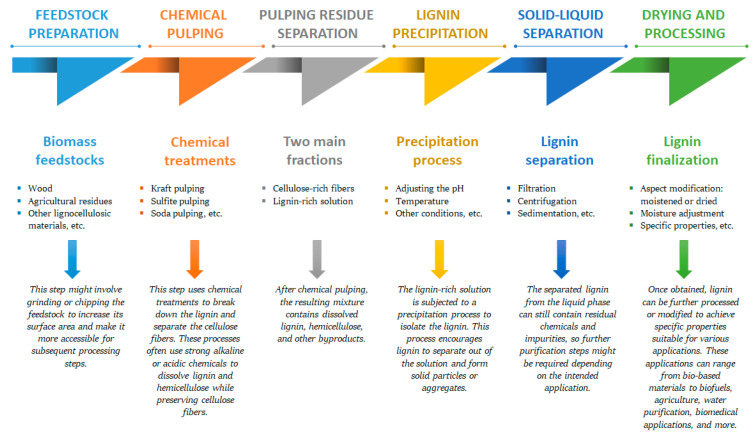
A typical manufacturing workflow for the processes of lignin biopolymer isolation from different biomass feedstocks.

**Figure 2 ijms-24-13493-f002:**
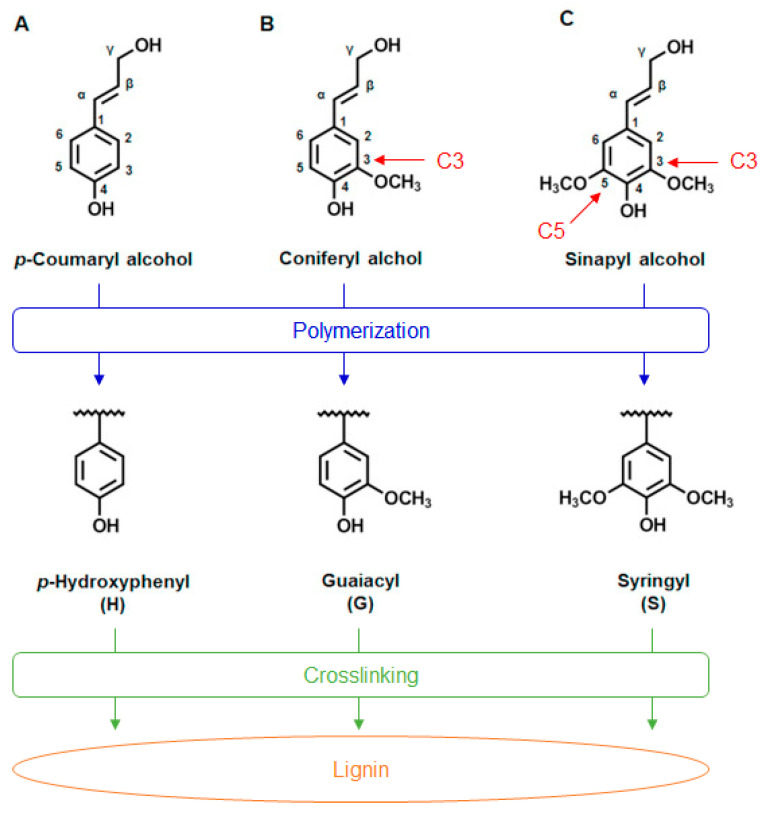
Chemical structures of three monolignols, which represent the precursors for the structural units in the lignin biopolymer: (**A**) p-coumaryl alcohol, which has no methoxy group; (**B**) coniferyl alcohol, which has one methoxy group at position C3; (**C**) sinapyl alcohol, which has two methoxy groups at positions C3 and C5. Below each chemical structure are the elemental monomers p-hydroxyphenyl (H), guaiacyl (G), and syringyl (S) (reproduced and adapted from Mayr et al., 2021 [58]; Copyright© 2021 MDPI under the terms of the Creative Commons Attribution 4.0 International License).

**Figure 3 ijms-24-13493-f003:**
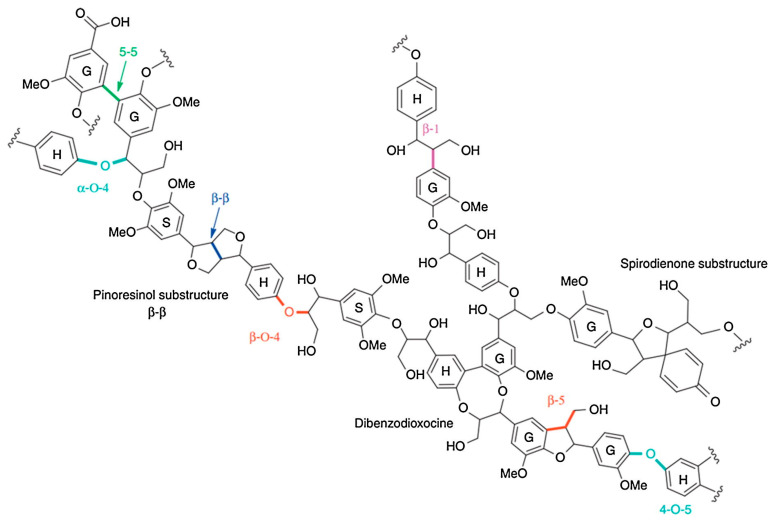
Example of lignin’s structure with the main linkage bonds (reprinted from Figueiredo et al., 2018 [62], with permission from Elsevier; published under license, Copyright© 2018 Elsevier Ltd.).

**Figure 5 ijms-24-13493-f005:**
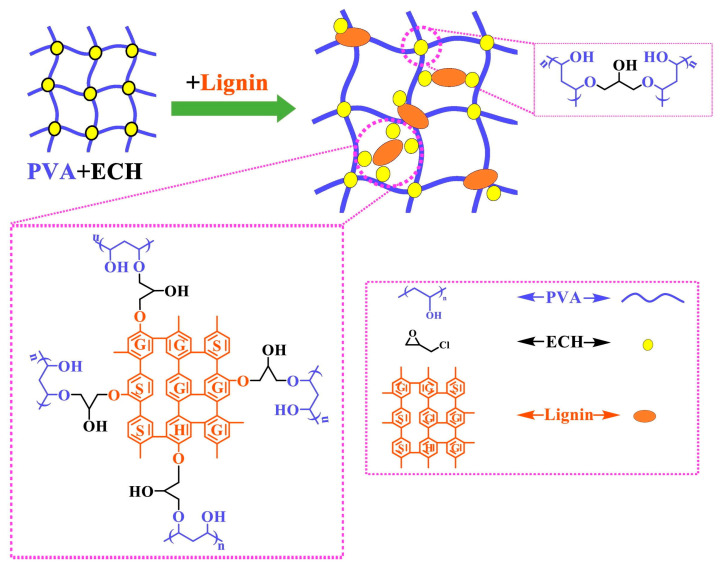
The structural mechanism of the lignin–PVA hydrogel: Through ECH, lignin–PVA and PVA–PVA might crosslink with one another to form the lignin–PVA hydrogel (reprinted from Wu et al., 2019 [110], with permission from Elsevier; published under license, Copyright© 2019 Elsevier Ltd.).

**Figure 6 ijms-24-13493-f006:**
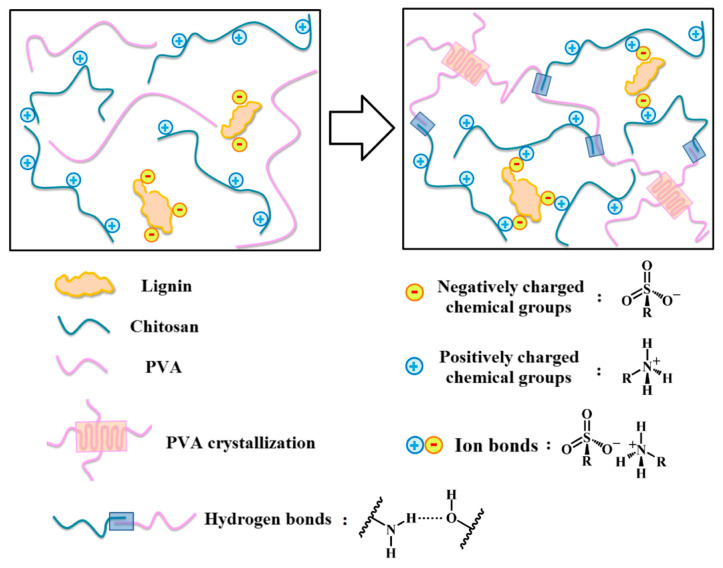
The crosslinking mechanism of lignin–chitosan–PVA hydrogel: active groups of lignin, such as hydroxyl, carboxyl, and sulfonic groups, can form hydrogen bonds (with groups in the chemical structures of chitosan and/or PVA) and ionic bonds (with groups in the chemical structures of chitosan and/or PVA) (reprinted from Zhang et al., 2019 [115], with permission from Elsevier; published under license, Copyright© 2019 Elsevier Ltd.).

**Figure 7 ijms-24-13493-f007:**
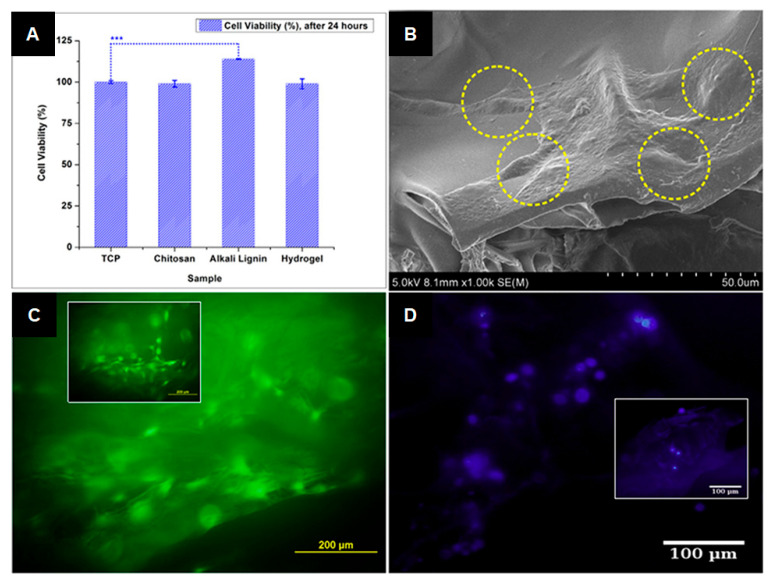
(**A**) Mesenchymal stem cell viability as a function of different materials (tissue culture plate (TCP), chitosan, alkali lignin, and chitosan–alkali lignin hydrogel). *** There was a significant difference (*p* < 0.05) between samples of alkali lignin and TCP. (**B**) Morphological image of the cell adhesion on chitosan–alkali lignin hydrogel (in yellow dotted circles). (**C**,**D**) FDA- and DAPI-stained fluorescence images of cell adhesion on chitosan–alkali lignin hydrogel, respectively (inset: fluorescence micrographs showing the cell adhesion on chitosan) (reprinted from Ravishankar et al., 2019 [23], with permission from Elsevier; published under license, Copyright© 2019 Elsevier Ltd.).

**Figure 8 ijms-24-13493-f008:**
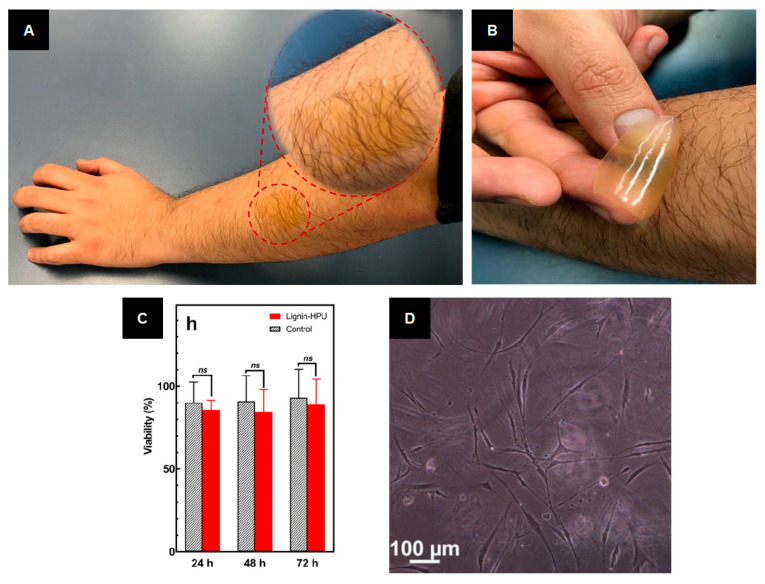
Application of hydrogels based on lignin and hydrophilic polyurethane: (**A**) Lignin–hydrophilic polyether-based polyurethane patch on arm. (**B**) The hydrogel film peeling off without any attached hair and without any pain. (**C**) Viability of human dermal fibroblast cells on lignin–hydrophilic polyether-based polyurethane film versus controls (no material) over three days (ns: No significant difference (*p* > 0.05) between lignin-HPU materials and controls). (**D**) Human dermal fibroblast cells on lignin–hydrophilic polyether-based polyurethane film (reproduced and adapted from Oveissi et al., 2018 [24], with permission from the American Chemical Society; Copyright© 2018 ACS Publications).

**Figure 9 ijms-24-13493-f009:**
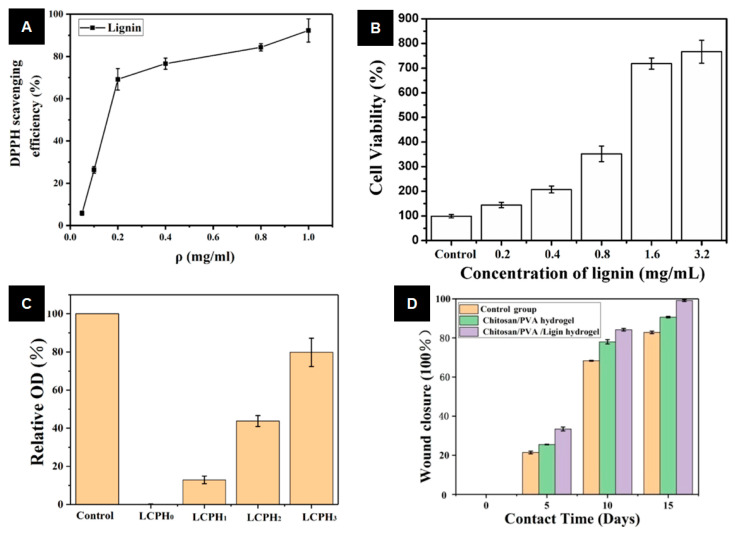
(**A**) Antioxidant properties of lignin. (**B**) Biological activity of lignin. (**C**) Antibacterial activity of lignin–chitosan–PVA hydrogels against *S. aureus*. (**D**) The wound recovery rate calculated from the wound recovery area treated with the control group (no dressing), chitosan–PVA hydrogel, and lignin–chitosan–PVA hydrogel dressing (reproduced and adapted from Zhang et al., 2019 [115], with permission from Elsevier; published under license, Copyright© 2019 Elsevier Ltd.).

**Figure 10 ijms-24-13493-f010:**
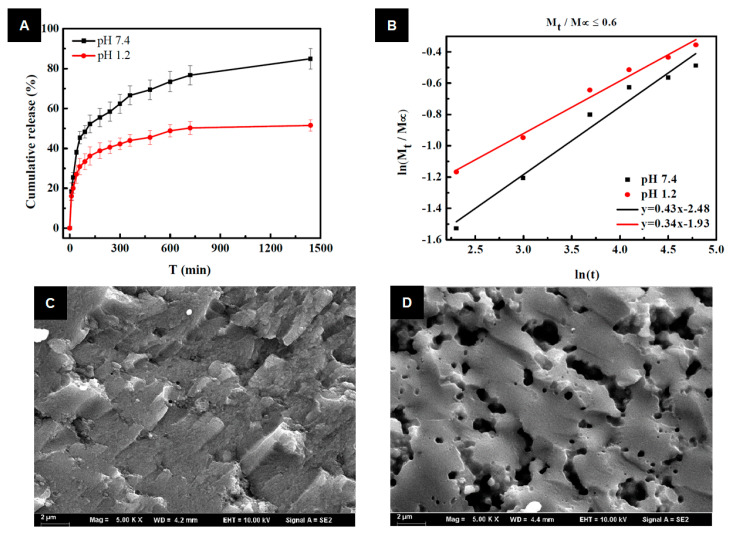
(**A**) The cumulative release curves of amoxicillin from sodium lignosulfonate−g−P(AA−co−PVP) hydrogels in simulated gastrointestinal fluids (simulated gastric fluids (pH = 1.2) and simulated intestinal fluids (pH = 7.4)) at 37 ± 0.5 °C. (**B**) Drug release dynamics of sodium lignosulfonate−g−P(AA−co−PVP) in simulated gastrointestinal fluid solution. (**C**) The partial enlarged detail of P(AA−co−PVP); (**D**) The partial enlarged detail of sodium lignosulfonate−g−P(AA−co−PVP) (reproduced and adapted from Wang et al., 2016 [25], with permission from the Royal Society of Chemistry; published under license, Copyright© 2016 RSC Publishing).

**Figure 11 ijms-24-13493-f011:**
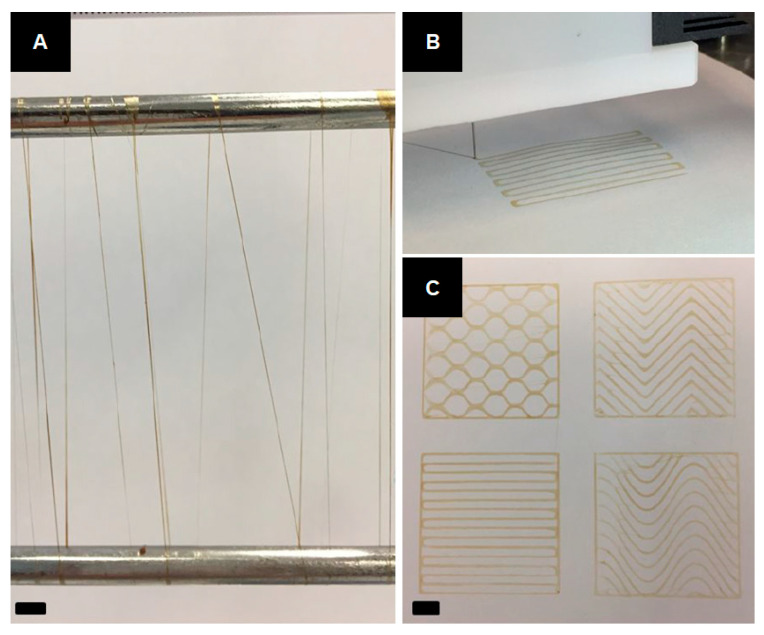
Application of hydrogels based on lignin and hydrophilic PU for 3D bioprinting applications: (**A**) Dry-spun hydrogel fibers. (**B**) Extruding hydrogel ink from the 3D printer nozzle. (**C**) Various patterns printed from hydrogel ink (reproduced and adapted from Oveissi et al., 2018 [24], with permission from the American Chemical Society; Copyright© 2018 ACS Publications).

**Figure 12 ijms-24-13493-f012:**
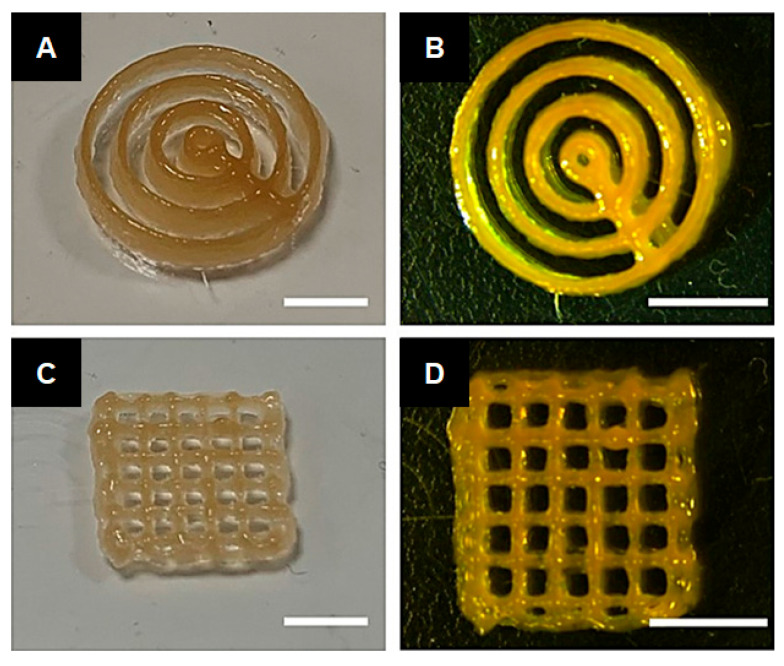
Images of the printed 3D structures based on the formulation of lignin–gellan gum hydrogels (bars = 5 mm): (**A**,**B**) Concentric cylindrical 3D structures. (**C**,**D**) Grid 3D structures (reproduced and adapted from Bonifacio et al., 2022 [126], with permission from Elsevier; published under license, Copyright© 2022 Elsevier Ltd.).

**Table 1 ijms-24-13493-t001:** Review of lignin-based hydrogels obtained by chemical or physical interactions as a function of lignin’s roles (i.e., crosslinked unit or crosslinking agent).

Crosslinking Type	Lignin Role	Matrix *	Crosslinker *	Ref.
Chemically crosslinked lignin-based hydrogels	Lignin as a crosslinked unit	Phenol–lignin–formaldehyde	Formaldehyde	[32]
		Lignin–cellulose	ECH	[33]
		Lignin amine	PEGDGE	[104]
		Acrylamide–PVA–graft lignin copolymers	NMBA	[105,106]
		Lignin–agarose	ECH	[107]
		Lignin–xanthan	ECH	[31,108]
		Acrylic acid–acrylamide–lignin	NMBA	[109]
		Lignin–PVA	ECH	[110]
		PVA–lignin epoxy	ECH	[8]
		Modified lignin–PVA	ECH	[111]
		Lignin–Mt–acrylic acid	NMBA	[112]
		Acrylic acid–OMt-grafted lignin	NEBA	[91]
	Lignin as a crosslinking agent	PVA	Aminated lignin	[71]
		PMVE/MA	Lignin and lignin–PEG	[103,113]
		Polymerized acrylic acid–PVP	Sodium lignosulfonate	[25]
Physically crosslinked lignin-based hydrogels	Lignin as a crosslinked unit	Lignin and hydrophilic PU	N/A	[24]
		Hydroxyethylcellulose–PVA	Borax	[114]
	Lignin as a crosslinking agent	Chitosan	Lignin	[23]
		Chitosan–PVA	Lignin	[115]

* Abbreviations: ECH: epichlorohydrin; PEGDGE: poly(ethylene glycol) diglycidylether; NMBA: N,N’-methylenebisacrylamide; NEBA: N,N’-ethylenebisacrylamide; PEG: poly(ethylene glycol); PVA: poly(vinyl alcohol); Mt: montmorillonite; OMt: organo-montmorillonite; PMVE/MA: poly(methyl vinyl ether-co-maleic acid); PU: polyurethane.

**Table 2 ijms-24-13493-t002:** Overview of lignin hydrogels in biomedical applications.

Lignin Hydrogels	Lignin (wt.%)	Biomedical Applications	Preparation	Hydrogel Properties	Ref.
Chitosan–alkali lignin	NDA ^1^	Tissue engineering and wound healing	Mixing an aqueous–acidic solution of chitosan with alkali lignin to form a physical hydrogel	Greater viscoelastic properties, good biocompatibility, and a conductive surface for cell attachment and growth	[23]
Lignin–PAAm	NDA ^1^	Broad range of applications in tissue engineering	Ultrasonic treatment for lignin nanoparticle dispersion. In situ free-radical polymerization for lignin–PAAm hydrogel	Excellent mechanical properties and non-toxicity	[121]
Lignin–alginate	3	Wide range of applications in tissue engineering and regenerative medicine	Exposure of an alginate–lignin–calcium carbonate aqueous–alkaline solution to pressurized carbon dioxide for hydrogel formation	Good cell adhesion properties without compromising the cell viability	[124]
Lignin–chitosan–PVA	10	Wound healing	Mixing of an aqueous–acidic solution of chitosan with lignin and PVA aqueous solution	The addition of lignin enhanced the mechanical strength, protein adsorption capacity, and cell proliferation properties of lignin–chitosan–PVA hydrogels	[115]
Lignin and hydrophilic PU	0–25	Wound healing and 3D bioprinting	Additional crosslinking of hydrophilic PU hydrogels with lignin by forming hydrogen bonds between the PU and the polar sites of lignin’s backbone	Improvement of the mechanical strength and processing ability of hydrophilic PU hydrogels. Good biocompatibility with primary human dermal fibroblasts. Possibility of scalable fabrication methods such as 3D printing, fiber spinning, and film casting	[24]
Lignin-grafted polyoxazoline-conjugated triazole	10	Wound healing	The hydrophilic polyoxazoline chain was grafted through ring-opening polymerization, and the copolymer was covalently modified with triazole	Prevented infection of the burn wound. Aided healing and the capacity as anti-inflammatory dressing material	[125]
PMVE/MA with lignin and lignin–PEG	10	Drug delivery	Lignin was combined with PMVE/MA and PEG to form a highly swellable hydrogel for the controlled release of hydrophobic curcumin	The hydrogel demonstrated logarithmic reductions in the adhesion of *S. aureus* and *P. mirabilis*	[113]
Lignin–xanthan–ECH	NDA ^1^	Drug delivery	Lignin–xanthan hydrogel using ECH as a crosslinking agent	The controlled release of hydrophilic bisoprolol fumarate for high blood pressure and heart failure treatments	[108]
Lignin–cellulose–ECH	25	Drug delivery	Lignin was mixed with cellulose and ECH to form a hydrogel	High swelling capacities are used for the release of polyphenols	[68]
Lignin-polymerized acrylic acid–PVP	NDA ^1^	Drug delivery	Sodium-lignosulfonate-grafted poly(acrylic acid-co-poly(vinyl pyrrolidone))	Hydrogel exhibited favorable pH sensitivity and controllable release behavior in vitro	[25]
Lignin–gellan gum	NDA ^1^	3D-bioprinted scaffold for cartilage repair	Blend of gellan gum and lignin to form a bioprintable hydrogel	Good rheological properties in terms of shear-thinning behavior and printability; the chondrogenic potential of the 3D structure was satisfactory	[126]
Lignin–cellulose–alginate	0–0.5	3D bioprinting	Spherical colloidal lignin particles were used to prepare lignin–cellulose–alginate nanocomposite bio-inks	Increasing the viscosity and improving the printability and shape stability of the composite hydrogels; no negative effect on cell viability	[127]
Esterified dealkaline lignin	NDA ^1^	3D bioprinting	Photopolymerization-based digital light processing with the addition of the co-initiator ethyl 4-(dimethylamino)benzoate	Dealkaline lignin was esterified to enhance its photoinitiation. Esterification of lignin enhances photoinitiation. Cell viability and proliferation improved	[128]

^1^ NDA: no data available.

## Data Availability

The data presented in this study are available within this manuscript.

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
