# Peer review of "Sustainable Biomass Lignin-Based Hydrogels: A Review on Properties, Formulation, and Biomedical Applications"

_ijms, 2023, doi:10.3390/ijms241713493_

Round 1

Reviewer 1 Report

Overall it is a well composed manuscript. Only a few edits are suggested as follows:

1) Figure 1 is poorly conceived. It is unclear what kind of pulping method was used to separate the cellulose fibers and lignin, and what kind of separation was done to isolate lignin from spent pulping liquor. The figure caption is also confusing (was formic acid used during pulping and adding water to the spent liquor helped to isolate the lignin? If so, what are the solid residues??).

It will be better to provide a general concept of lignin fractionation from different feedstocks using a generic chemical pulping and precipitation method. Clip arts and a process flow diagram could be used.

2) Figure 4 does not explain physical or chemical crosslinking to form the macromolecular network. Please somehow incorporate these concepts.

3) What is the difference between figure 5 and figure 6? They are both essentially explaining the same concept. Figure 6 provides a better explanation for lignin modification and chemical crosslinking with PVA and ECH. Therefore, I recommend to keep this figure (6) and remove the superfluous (and low resolution) figure 5.

Minor edits and attention to detail is required. E.g.:

1) Bacteria names, like S. aureus, must be in italics.

2) Figure captions must be self-sufficient. Please exclude abbreviations and provide full names of HPU (Figure 9), SLS, SGF, and SIF (Figure 11).

3) Sometimes the language gets too complicated like Figure 1 caption and Figure 11 caption. Please try to simplify.

Author Response

Please find the response to the reviewer's comments attached.

Reviewer 2 Report

In the manuscript by Fatimi et al., the authors provide a comprehensive review of lignin-based hydrogel materials and their biomedical applications. The authors begin by highlighting how lignin materials are prepared from natural sources, then offer an overview of the synthesis of lignin-based hydrogels. They further expand on the biomedical applications of these hydrogel materials, including tissue engineering, wound healing, drug delivery, and 3D printing, offering insightful projections for the future development of this field. Despite the existing wealth of reviews on lignin materials, as the authors themselves mention, a dedicated focus on lignin-based hydrogels and their specific biomedical applications is still lacking. Therefore, I recommend the publication of this manuscript once the following comments have been thoroughly addressed:

1.       In Section 2.3.4, the authors introduce other critical properties of lignin for biomedical applications. However, the organization of this subsection could be improved. A summary paragraph at the beginning, outlining these properties at a high level before discussing each in detail, would be helpful for reader comprehension.

2.       Section 3.1 is a long paragraph introducing the basic concepts of hydrogels. While I appreciate the authors' intentions of providing a comprehensive overview for readers less familiar with these concepts, some descriptions less relevant to the manuscript's scope, such as stimuli responsiveness, could be trimmed to maintain focus and conciseness.

3.       The authors' claim in Lines 294-296 that PHEMA was the first synthesized hydrogel is potentially incorrect. Although PHEMA marked a significant turning point in the production of hydrogel materials, the first reported hydrogel was Ivalon, a crosslinked network made from PVA and formaldehyde (Grindlay JH, Clagett OT. Proc Staff Meet Mayo Clin. 1949, 24, 538). The authors should confirm the accuracy of this statement.

4.       It's good to see that the authors have included 3D printing as one of the applications of lignin-based hydrogel materials. While they describe several representative examples, the omission of seminal works by Lewis and coworkers from Harvard is a significant oversight. As pioneers in the field of DIW 3D printing, Lewis and her team deserve due credit in the discussion of 3D printing concepts.

The language is largely free of grammar issues but could improve in terms of conciseness. Please see the comments above for more details. Thank you.

Author Response

(The authors gave the same response as above.)

Reviewer 3 Report

The review "Sustainable biomass lignin-based hydrogels for biomedical applications: A review" includes extensive information about the richness of the chemistry of natural lignin and the possibilities of its use. The list of references analyzed by the authors is very significant and includes 167 references, including modern reviews by other authors. The advantages of the review include the authors' attempt to describe the rich chemistry of natural lignin, to illustrate the complexity of the biopolymer structure with pictures, and an attempt to find the reasons that provide lignin with the constancy of scientific attention and the desire to develop its capabilities in various industries. But the shortcomings of this review are also obvious to me. In particular, there is too much information, and the name promised to reveal the nature of lignin-based hydrogels. Therefore, my specific comments boil down to the need to revise the manuscript in order to specify all the information in accordance with the title.

Specific remarks:

1. Reduce the volume of the review and the number of references, bring it in line with the name "lignin-based hydrogels". Otherwise, you need to change the name of the review.

2. Lignin is known to be the third most abundant polymer in plants [101, the so-called first review of lignin-based hydrogels]. Why does this manuscript call him the second?

3. It is known that “Lignin is widely used as a raw material for the synthesis of bio-based hydrogels due to its hydrophilic nature…. Methods for the synthesis of lignin-based hydrogels include interpenetrating polymer network, cross-linking copolymerization, atom transfer radical polymerization, and reversible polymerization with the addition-fragmentation chain transfer [101, the so-called first review on lignin-based hydrogels]. This manuscript lacks a clear idea of the mechanism for creating hydrogels based on the polymeric nature of lignin.

4. The abstract does not reflect the title of the review and does not reveal the mechanism of formation of hydrogels based on natural lignin. And most importantly, there is no information in the annotation that "without copolymerization, it is impossible to obtain hydrogels using lignin." But there is information that the manuscript covers various methods for isolating lignin.

5. The text needs to be redone to avoid misinforming readers about the role of lignin as a "backbone in hydrogels". For example, lines 517-530: the preparation of a hydrogel is described, in which lignin acts as a plasticizer, and not a polymer "base", and at the end of the description, the authors conclude that "Thus, the developed lignin-based hydrogel was more suitable ..." .

6. The most popular section of the IJMS edition “4. Biomedical applications of lignin-based hydrogels” argues that the promising products discussed in it owe their sought-after properties to lignin. But this is not true. The section describes products that are complex in composition, their unique properties are provided precisely by the complexity of their composition. I think that table 3 misinforms readers by its title. What is the mass contribution of lignin in these products?

Author Response

(The authors gave the same response as above.)

Round 2

Reviewer 3 Report

I followed all the changes that the review went through during the peer review process: from the title change, the rewritten abstract, changes in the text, tables, supplementary, including the reduction in the number of references. I believe that the review "Sustainable Biomass Lignin-based Hydrogels: A Review on Properties, Formulation, and Biomedical Applications" can be published as presented.